# Effects of Some Hill Reaction-Inhibiting Herbicides on Nitrous Oxide Emission from Nitrogen-Input Farming Soil

**Yuta Takatsu, Sharon Y. L. Lau [†] , Li Li and Yasuyuki Hashidoko ***

Research Faculty of Agriculture, Hokkaido University, Kita 9 Nishi 9, Kita-ku, Sapporo 060-8589, Japan;
record@eis.hokudai.ac.jp (Y.T.); lauyuling@gmail.com (S.Y.L.L.); blueocean0301@gmail.com (L.L.)

* Correspondence: yasu-h@abs.agr.hokudai.ac.jp

† Present address: Sarawak Tropical Peat Research Institute, 94300 Kota Samarahan, Sarawak, Malaysia.

**Abstract:** Nitrous oxide ($N_2O$) emission-suppressing activity of some electron-transport inhibitors of the Hill reaction system was investigated. The Hill reaction inhibitors—paraquat, isouron, bromacil, diquat, and simazine—all of which have been or are currently being used as herbicides in farming activity are expected to inhibit the electron-transporting pathways of nitrate respiration in denitrifying bacteria. Using $N_2O$-emitting soil bed (5.0 g of fresh weight) from a continuously manured Andisol corn farmland in Hokkaido, Japan, which was autoclaved and further supplemented with an active $N_2O$-emitter, *Pseudomonas* sp. 5CFM15-6D, and 1 mL of 100 mM $NH_4NO_3$ or $(NH_4)_2SO_4$ solution as the sole nitrogen source (final concentration, 0.2 mM) in a 30 mL gas-chromatography vial, the effects of the five herbicides on $N_2O$ emission were examined. Paraquat and isouron (each at 50 μM) showed a statistically significant suppression of $N_2O$ emission in both the nitrification and the denitrification processes after a 7-day-incubation, whereas diquat at the same concentration accelerated $N_2O$ emission in the presence of $NO_3^-$. These results suggest that paraquat and isouron inhibited both the nitrification and the denitrification processes for $N_2O$ generation, or its upstream stages, whereas diquat specifically inhibited $N_2O$ reductase, an enzyme that catalyzes the reduction of $N_2O$ to $N_2$ gas. Incomplete denitrifiers are the key players in the potent emission of $N_2O$ from Andisol corn farmland soil because of the missing *nosZ* gene. The electron relay system-inhibiting herbicides—paraquat and isouron—possibly contribute to the prevention of denitrification-induced nitrogen loss from the farming soil.

**Keywords:** electron-transport inhibiting herbicide; Hill reaction inhibitors; denitrification inhibition; Andisol farmland soil; $N_2O$ emission suppression

## 1. Introduction

Nitrogen, an essential element required for plants and other living creatures, is mainly provided in the form of amino acids (as degraded protein in soil organic matters), ammonium ($NH_4^+$, stable in soil), or nitrate ($NO_3^-$, easily lost from soil through leaching and denitrification) [1,2]. Plants or fungi inhabiting nitrogen-deficient soil acquire available nitrogen as follows: (1) Establishing symbiosis with free-living nitrogen fixing bacteria, (2) decomposing organic substances actively, or (3) suppressing nitrification of bacteria or archaea in soil, known as biological nitrification inhibition (BNI) [3,4]. *Brachiaria humidicola* and *Sorghum bicolor* of the family Poaceae (also known as Gramineae), release diterpene (brachialactone) and paraquinone derivatives (soligoquinones), respectively, from the plant roots as a BNI mechanism [5,6]. This mechanism involves the suppression of nitrogen loss from soil by the selective inhibition of ammonia oxidase in nitrifiers [4,6]. Whereas, denitrification is the most

important process associated with nitrogen loss in natural or farming soil [7,8]. To the best of our knowledge, natural products or synthesized chemical agents that can suppress this process, towards the reduction of nitrate respiration, are rarely known [9,10].

$N_2O$ emission in farmland is attributable to human activities and it has a huge effect on global warming, because $N_2O$ is a potent greenhouse gas which accounts for 6–8% of global warming [11]. $N_2O$ is also a major factor affecting ozone depletion [12]. In acidic and fertilized soil utilized for farming activity, the final denitrification process, which reduces $N_2O$ to $N_2$ through the catalytic reduction by $N_2O$ reductase (NosZ), is often inhibited at the level of gene (*nosZ*) transcription [13,14]. In addition, lowered enzymatic activity in the acidic region can also inhibit the final denitrification process [15]. Hence, acidic soils such as acidic peat soils, acid-sulphate soil, coniferous forest bed soils, excess ammonium sulphate-containing-soil, tropical red soil, or volcanic ash soil, often become strong $N_2O$ emission spots [16–19].

Nevertheless, some agricultural farmlands reclaimed from peat swampy forests, including oil palm plantation soil in Sarawak, Malaysia, showed relatively low $N_2O$ emission despite the large addition of nitrogen and mineral fertilizers to the soil [20]. Similar woody peatland in Sumatra, Indonesia, which was converted to acacia plantation, particularly those in the mature plantation soil, suppressed $N_2O$ emission despite the N-fertilizer input [21]. Using our culture-based $N_2O$ emission assay, we came to the conclusion that practical $N_2O$ flux from the soil was suppressed due to the introduction of an acceptable amount of paraquat to the plantation [20]. If any herbicide selectively inhibited nitrate respiration process treating $NO_x$ as an electron acceptor instead of $O_2$, the herbicide may block the reduction process of denitrification to suppress $N_2O$ emission, same as paraquat.

Thus, some Hill reaction-inhibiting herbicides showed potentials to suppress $N_2O$ emission of nitrifiers and denitrifiers in the soil. According to the results of our bioassay, we show some evidence for Tollefson's short comments on the hidden effect of chemical pesticides [22,23].

## 2. Materials and Methods

### 2.1. Chemicals

Five electron transport inhibiting herbicidal compounds, methyl viologen dichloride (paraquat **1**; reagent grade), 3-(5-(tert-butyl)isoxazol-3-yl)-1,1-dimethylurea (isouron **2**; reagent grade), 5-bromo-3-*sec*-butyl-6-methyluracil (bromacil **3**, reagent grade), 1,1-ethylene-2,2'-bipyridinium (diquat **4**; reagent grade) [24], and 2-chloro-4,6-bis(ethylamino)-*S*-triazine (simazine **5**; reagent grade) were purchased from Wako (Osaka, Japan). Tropolone (**6**) was also a product from Wako (Figure 1). All the herbicides used are Hill reaction inhibitors preventing electron transport in the photosynthetic electron relay system [25]. In the soil incubation assay, $N_2O$-emitting soil of Andisol corn farmland, a recognized $N_2O$ emission hotspot [21,26,27], was exposed to 50 μM of each test compound.

### 2.2. $N_2O$ Emitting Soil for $N_2O$ Emission Assay

Andisol at 0–15 cm of depth was collected from the fertilized dent corn farmland at Hokkaido University Shizunai Experimental Livestock Farm in Hokkaido, Japan (42°26′ N, 142°28′ E) in late April 2016, before tillage and at the first fertilization. *Pseudomonas* species, which are incomplete denitrifiers and $N_2O$ emitters, have been isolated from the soil [21]. In addition, indirect evidence from another report shows that some unculturable soil microorganisms highly contribute to the active $N_2O$ emission from the soil in early spring time [26].

**Figure 1.** Test compounds.

### 2.3. Culture-Based N₂O-Emission Inhibition Assay Using Hill Reaction Inhibitors towards an Incomplete Denitrifier Isolated from the Andisol

Winogradsky's mineral solution-based 0.3% gellan gum soft gel culture (10 mL) was supplemented with 0.5 mg/mL sucrose and 1.44 mg/mL (14 mM) $KNO_3$ as carbon and excessive nitrogen sources, respectively. The culture was adjusted to pH 6.0 and sealed in gas-chromatography vials (Nichiden-Rika Glass Co., Kobe, Japan). After being autoclaved (121 °C for 15 min), 890 μL of *Pseudomonas* sp. 05CFM15-6D cell suspension ($OD_{660}$ = 0.1) and 110 μL of 1.0 mM chemical solution (as 10% aq. DMSO) were added to the medium (finally 1.0% aq. DMSO). Hence, the headspace volume was 21.5 mL (27 mm of inner diameter, 490 $mm^2$ of opening area). Because the denitrification events are effective respiration processors under anaerobic conditions where $NO_3^-$ is an electron acceptor, some electron transport-inhibiting herbicides isouron (**2**), bromacil (**3**), diquat (**4**), and simazine (**5**) were tested at 10 μM, along with the same concentration of tropolone (**6**). Incubation was done at 25 °C in the dark for 2 weeks.

### 2.4. Soil Bed-Based N₂O Emission with Sucrose Supplementation

For the $N_2O$ production assay in a soil bed, 5.0 g of the raw soil was put into a 30 mL gas chromatography vial, and 100 μL of 100 mM $NH_4NO_3$ solution was added to the soil at a final concentration of 2 mM nitrogen sources ($NH_4^+$ and $NO_3^-$). To induce a stable $N_2O$ emission by incomplete denitrifiers, sucrose was added to the soil as an additional carbon source [21,27,28]—(**a**) 100 μL of 100 mg/mL sucrose solution was added to make a final concentration of 0.19% (*w/w*). Water-filled weight-base measurement of headspace volume was performed as described in the literature [29], the headspace was 27.3 mL (*n* = 10). (**b**) Alternative sucrose content (0.01–1.0%) was tested [28]. Incubation was done at 25 °C in the dark for 7 days.

### 2.5. Soil Bed-Based N₂O Emission Inhibition Assay

In the $N_2O$ emission suppression assay, a *Pseudomonas* sp. isolated from Shinhidaka Andisol corn farm soil as an incomplete denitrifier was used [21]. $N_2O$ emission from 5 g of the soil bed in a gas-chromatography vial and its suppression by Hill reaction-inhibiting herbicides were tested. We used the autoclaved soil inoculated with *Pseudomonas* sp. 05CFM15-6D and minimized 0.05% sucrose relatively stable for the $N_2O$ emission, according to the results of Section 2.4. To the autoclaved soils containing inorganic nitrogen salt (0.2 mM $NH_4NO_3$ or 0.2 mM $(NH_4)_2SO_4$) as possible substrates of

$N_2O$, and 0.05% sucrose as the carbon source, 50 μM of each Hill reaction-inhibiting herbicide and 890 μl of the bacterial cell suspension ($OD_{660}$, 0.1) were added, and the mixture was vortexed before incubation. $N_2O$ accumulated in the headspace (27.3 mL) was analyzed quantitatively by using a gas chromatography instrument as described next.

### 2.6. Quantitative Measurement of $N_2O$ in Headspace

The level of $N_2O$ in the headspace gas of the cultured vials with inoculates was measured using electron capture detector (ECD)-gas chromatography (GC-14B equipped with ECD-2014, Shimadzu, Kyoto, Japan). The gas chromatograph was equipped with an ECD maintained at 340 °C using a 1 m Porapak N column (Waters, Milford, MS, USA) maintained at 60 °C, with argon supplemented with 5% $CH_4$ as the carrier gas. After 2 weeks incubation, a portion of the headspace gas (50 μL–1.0 mL) was analyzed by the gas chromatography. For the quantification of $N_2O$ gas, a standard curve was made using an absolute calibration method [27]. A series of concentrations (0, 0.498, 4.98, 49.8, and 498 ng/mL nitrogen gas) of standard $N_2O$ gas were injected as a 1.0 mL volume for generating the standard curve to cross the origin. In this quantification of $N_2O$, a high precision is needed to monitor atmospheric levels of $N_2O$ with a detection limit of 0.05 ng/L (100 ppb).

### 2.7. Effects of Hill Reaction-Inhibiting Herbicides on $N_2O$ Quenching Chitinophaga

The effect of paraquat (**1**), diquat (**4**), and an iron-chelator tropolone (**6**), were selectively investigated on the $N_2O$ quenching effect of a *Chitinophaga* that was isolated from the soil of Andisol corn farmland in Shinhidaka, Hokkaido, Japan, using an $N_2O$ quenching assay system [30]. A 110 μL aliquot of a 10 mM sample solution dissolved in dimethylsulfoxide (DMSO) was added to Winogradsky's mineral solution-based 0.5% gellan gum soft gel culture (10 mL) supplemented with 0.5 mg/mL sucrose and 0.7 mg/mL aspartic acid as carbon and nitrogen sources, respectively. The culture was adjusted to pH 6.0 and sealed in gas-chromatography vials.

The gellan gum gel bed contained 100 μM of the test compound. The sample in DMSO as 10- and 100-fold diluted with Milli-Q to prepare 10 and 1 μM test compound-containing media. To all of the assay media, 890 μL of $N_2O$ quenching *Chitinophaga* cell suspension was inoculated. In the blank medium, 890 μL Milli-Q was added instead. Finally, $N_2O$ standard gas (standard $N_2O$ gas, GL Sciences, Tokyo, Japan) was injected with a gas-tight syringe into the headspace (21.6 mL) of the gas-chromatography vials to make a final concentration of 12,000 ppmv. The culture in the assay vial was incubated at 25 °C for 6 days in the dark. After 3-day- and 6-day-incubation, the concentration of the remaining $N_2O$ gas in the headspace was quantified by ECD-gas-chromatography in comparison with the culture vials to which no chemical was added, or the blank.

### 2.8. Statistical Analysis

The cumulative $N_2O$ emissions were expressed as an arithmetic mean and standard deviation (±SD). Statistical analyses were done by Student *t*-tests.

## 3. Results

### 3.1. Acceleration of $N_2O$ Emission in Soil Bed Culture Supplemented with Nitrogen Substrate and Sucrose

As the Winogradsky's mineral solution-based gellan gum bed, supplementation of nitrogen substrates, particularly with 2 mM $NO_3^-$, and carbon source such as 0.2% sucrose resulted in the active acceleration of $N_2O$ emission from the soil bed in the gas-chromatography vial (Figure 2) [28]. Although 0.01–1.0% sucrose content was also tested in parallel, $N_2O$ emission from the soil bed culture was unstable, including control (0% sucrose), showing that the soil is thus far from a homogeneous material.

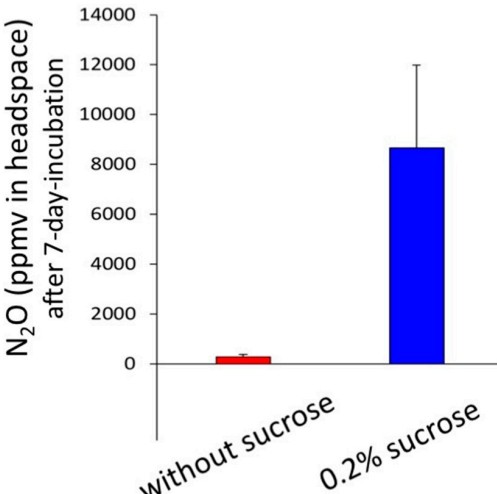

**Figure 2.** Acceleration of $N_2O$ emission from Andisol soil bed by sucrose supplementation. Effect of 0.2% sucrose supplementation on $N_2O$ emission from raw soil bed (5.0 g) from the Andisol corn farm placed in a gas-chromatography vial. Bars are standard deviation ($\pm$ SD, *n* = 5). Soils were also supplemented with 0.2 mM $NH_4NO_3$.

### 3.2. Suppression of $N_2O$ Emission from Soft Gel Medium Adding Hill Reaction-Inhibiting Herbicides

Among the herbicides (**2**–**5**) and tropolone (**6**), only diquat (**4**) showed a clear $N_2O$-emission-suppressing effect at 10 μM (Figure 3). After the 1-week-incubation, diquat (**4**) reduced $N_2O$ emission into less than a half, 106 ppmv (equivalent to 68 μg/mL medium/d) (cf. 276 ppmv in control, equivalent to 117 μg/mL medium/d). At 2-week-incubation, emitted $N_2O$ level in the cultured medium treated with 10 μM diquat was 161 ppmv (103 μg/mL medium/d) (cf. 320 ppmv in control, 205 μg/mL medium/d). Conversely, other chemical compounds tested at 10 μM did not show any inhibitory effect on $N_2O$ emission by *Pseudomonas* sp. 05CFM15-6D cultured in the soft gel medium.

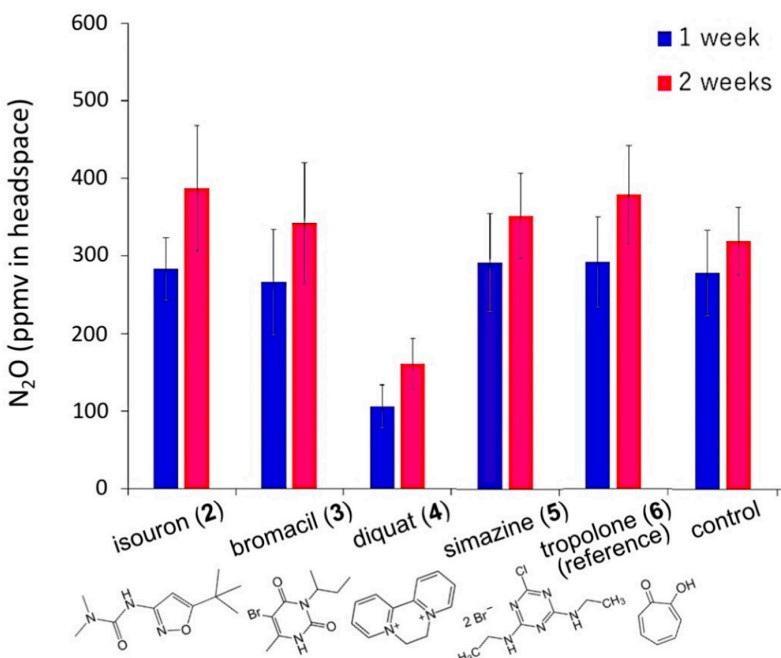

**Figure 3.** Suppression of $N_2O$ emission from Winogradsky's medium-based gellan gum bed inoculated with an incomplete denitrifier *Pseudomonas* sp. by the addition of Hill reaction-inhibiting herbicides. Ingredients of the medium are shown in Materials and Methods. Bars are standard deviation ($\pm$ SD, *n* = 3). Control contained 0.1% DMSO in the soft gel medium.

### 3.3. Suppression of N₂O Emission from an Autoclaved Soil of N₂O Emission Hotspot Andisol Followed by Inoculation with Pseudomonas sp. 05CFM15-6D

Approximately 195 ppmv level $N_2O$ (251 ng/g soil bed/d) was emitted from the $NH_4NO_3$-supplemented soil bed. In addition, approximately 340 ppmv $N_2O$ (equivalent to 437 ng/d/g soil bed) was emitted from the soil bed supplemented with 0.2 mM $(NH_4)_2SO_4$. Paraquat (**1**) at a final concentration of 50 μM showed a remarkable suppressing effect on $N_2O$ emission from the soil bed supplemented with 0.2 mM $(NH_4)_2SO_4$, in which the average concentration of $N_2O$ in the headspace was 58 ppmv (75 ng/d/g soil bed). Conversely, at the same concentration of paraquat, there was less suppression of $N_2O$ emission (97 ppmv, equivalent to 124 ng/d/g soil bed) from the 0.2 mM $NH_4NO_3^-$ supplemented soil bed. Isouron (**2**) also showed a similar inhibitory activity against $N_2O$ emission from the soil bed [98 ppmv (126 ng/d/g soil bed) and 127 ppmv (163 ng/d/g soil bed) for soil beds supplemented with 0.2 mM $(NH_4)_2SO_4$ and 0.2 mM $NH_4NO_3$, respectively]. Diquat (**4**) did not show any significant $N_2O$ emission suppression. Similarly, bromacil (**3**) and simazine (**5**) did not inhibit $N_2O$ emission. Although tropolone (**6**) is not a Hill reaction inhibitor but a potent iron-chelator [31], this iron-chelating compound was also tested as a reference compound. However, it did not show any suppressing effect on $N_2O$ emission (Figure 4).

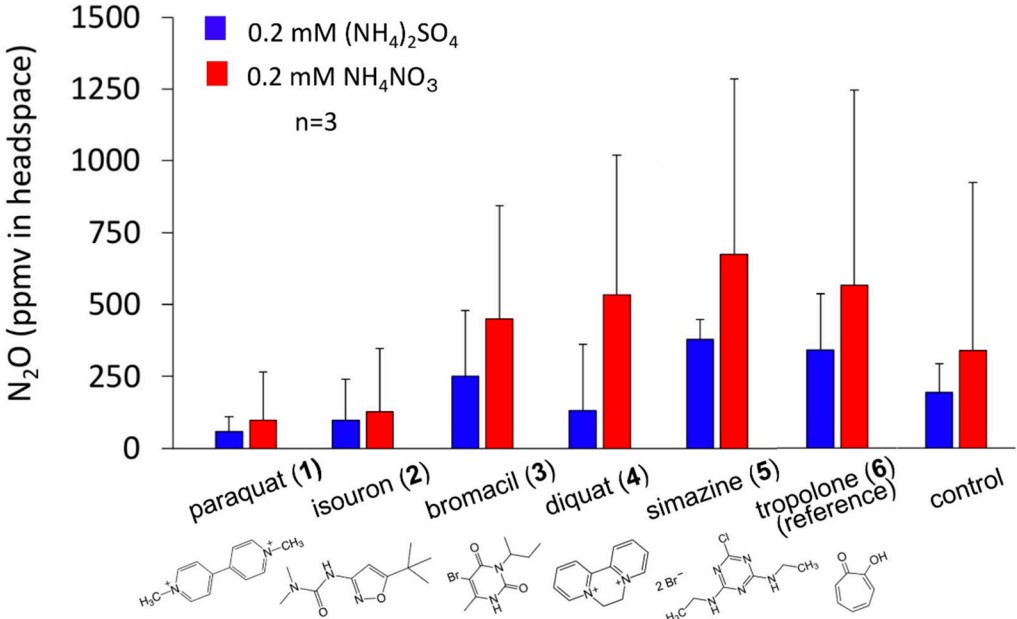

**Figure 4.** Suppression of $N_2O$ emission from Andisol soil bed by the addition of the Hill reaction-inhibiting herbicides. Incubation was performed for seven days. Bars are standard deviation (± SD, *n* = 3).

### 3.4. Suppressing Action of the Herbicides on Actively N₂O Quenching Chitinophaga Isolated from an Andisol Corn Farm Soil

Unlike $N_2O$ emission, the $N_2O$ quenching effect of the *Chitinophaga* bacterium was not inhibited by 1, 10, and 100 μM paraquat (**1**). Diquat (**4**), a bipyridylium-type Hill reaction-inhibiting herbicide, at a final concentration of 1 or 10 μM did not suppress $N_2O$ quenching, but 100 μM of **4** reduced the effectiveness of $N_2O$ quenching by *Chitinophaga* after the 3-day-incubation, with 60% of the $N_2O$ remaining in the headspace. Whereas, in all the vials at the three different concentrations, $N_2O$ was quenched to almost zero level at day 6. At a final concentration of 100 μM, treatment with tropolone (**6**) resulted in nearly 80% inhibition of the $N_2O$ quenching, although no effect was observed at 1 or 10 μM. This suppression of $N_2O$ quenching continued for 6 days, and at day 6, 40% $N_2O$ remained in the headspace (Figure 5).

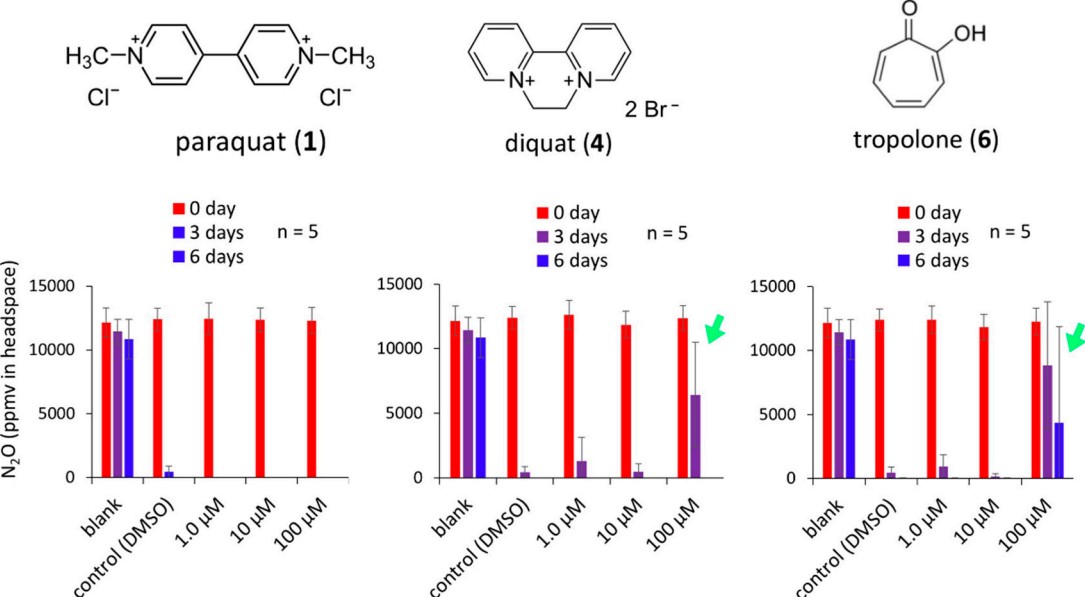

**Figure 5.** Effect of bipyridylium cation-type Hill reaction-inhibiting herbicides, paraquat, and diquat, on an N$_2$O quencher, *Chitinophaga* strain isolated from Andisol corn farming soil (green arrows). Bars are standard deviation (±SD, *n* = 5).

## 4. Discussion

The results of alternative sucrose supplementation to the raw bulk soil, showing no direct proportion to N$_2$O emission (Figure 2), suggested that soil microbial ecosystems in the bulk soil is consisted of diverse community members. Among them, some may show a faster response to sucrose rather than the incomplete denitrifiers. Hence, the soil, that is far from a homogeneous material, probably showed such an alternative response of N$_2$O emission to the carbon source added.

On the other hand, N$_2$O emission-inhibiting effect in the Winogradsky's mineral solution-based gellan gum soft gel medium was observed only in diquat (**4**) among the herbicides (**2**–**5**) and tropolone (**6**) tested at 10 μM (Figure 3). Using gellan gum gel bed for the culture-based N$_2$O emission assay, paraquat (**1**) and diquat (**4**), both of which are bipyridylium cation-type Hill reaction inhibitors, showed clear suppression of N$_2$O emission by the incomplete denitrifiers of *Pseudomonas* species [21]. Other chemicals (**2**, **3**, **5**, and **6**) did not show any inhibitory effect on N$_2$O emission at 10 μM, indicating that pyridilium cation moiety is necessary for denitrification inhibition.

In an autoclaved soil bed, however, two electron-transport inhibiting herbicides, paraquat (**1**) and isouron (**2**) at a final concentration of 50 μM suppressed N$_2$O generation from the 2 mM NH$_4$NO$_3{}^-$ supplemented model soil bed of fertilized Andisol farmland bulk soil (Figure 4). However, three of the electron-transport inhibiting herbicides (**3**–**5**), including diquat (**4**), did not show any potent suppressing effect on N$_2$O emission. Despite a similar bipyridylium cation structure with **1**, diquat (**4**) did not show any inhibiting activity against N$_2$O emission in the soil beds. Thus, N$_2$O-producible incomplete denitrification with NO$_3{}^-$ as the substrate was inhibited by **1** and **2** only. These clear differences in the suppressing effects of different Hill reaction inhibitors on the emission of N$_2$O from the soil bed culture suggest a specific inhibition of certain oxidoreductase, highly associated with the denitrification process and/or the nitrification process [20]. Conversely, the results shown in Figure 4 probably indicate that not only incomplete denitrifiers but also thermo-tolerant nitrifier is another player in N$_2$O emission in the farm soil. In the autoclaved soil, *amoA*-harboring thermo-tolerant archaea may be surviving and emerging as major nitrifiers in the soil bed [32].

In contrast, the inhibitory activity of 100 μM diquat (**4**) on N$_2$O quenching by *Chitinophaga* may indicate that N$_2$O production and N$_2$O quenching in the soil are separable responses [33]. Oxidoreductases associated with inorganic nitrogen metabolism are often specific to a Hill reaction

inhibiting herbicide. Tropolone (**6**), which inactivates iron-containing heme-dependent cytochrome c or iron-sulfur-cluster containing oxidoreductases showed a relatively high suppression of $N_2O$ quenching by *Chitinophaga*.

Thus, the current data demonstrates the roles of some Hill reaction-inhibiting or other herbicides in the suppression of $N_2O$ flux from agricultural farming soil [22,23]. This implication is important because bifunctional herbicides that can effectively prevent the loss of nitrogen by denitrification as well as control the growth of weed can possibly be developed with a molecular design and bio-rational screening systems [9,10,34]. Thus far to our knowledge, the mechanism and mode of action of biological denitrification inhibition are rarely known, but some reliable reports have been published [35,36].

## 5. Conclusions

In this study, we demonstrated that a 50 μM level of two electron transport-inhibiting (Hill reaction-inhibiting) herbicides, paraquat (**1**), and isouron (**2**), which have been approved as regal herbicides, showed potent suppression of $N_2O$ emission from the farm soil bed modified for fertilized conditions. This result implied that herbicides **1** and **2** positively contributed to environmental sustainability via suppression of nitrogen loss and $N_2O$ emission from fertilized soils. However, three of the herbicides (**3–5**) did not show any suppressing effect on $N_2O$ emission. This suggests that some electron-transport inhibiting chemicals are associated with selective steps of inorganic nitrogen metabolism. This specific inhibition targeting certain oxidoreductases, particularly the ones associated with the denitrification process, may provide new approaches that can be used to suppress $N_2O$ flux from agricultural soil and more importantly to prevent the loss of nitrogen from fertilized soil by denitrification.

**Author Contributions:** Conceptualization, Y.H.; Data curation, Y.T. and Y.H.; Funding acquisition, Y.H.; Investigation, Y.T., S.Y.L.L. and L.L.; Methodology, Y.T., S.Y.L.L., L.L. and Y.H.; Project administration, Y.H.; Resources, Y.T.; Validation, Y.H.; Writing—original draft, Y.T. and Y.H.; Writing—review & editing, Y.H.

**Funding:** This research was funded by Grant-in-Aid A (26252058) and B (26304042) to Yasuyuki Hashidoko.

**Conflicts of Interest:** The authors declare no conflict of interest.

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
