# Peer review of "Effects of Some Hill Reaction-Inhibiting Herbicides on Nitrous Oxide Emission from Nitrogen-Input Farming Soil"

_applsci, doi:10.3390/app9091903_

Reviewer 1 Report

Dear authors,

 Authors tested the impact of several kind of herbicides on nitrous oxide emissions from Andisol samples. The data are precious. Basically I agree with the publication of the manuscript, but several revision could be required. First, I encourage authors to revise the method parts in the manuscript. Throughfall the manuscript, it was difficult to follow the method of the present study. 2-2. Quantitative measurement of N2O in headspace should be presented after the description of incubation study. Probably, Yanai et al. (2007) is helpful to revise the manuscript. They performed an incubation study to test the impact of biochar addition on N2O emssions.

Yanai Y., Toyota K. & Okazaki M. (2007). Effects of charcoal addition on N2O emissions from soil resulting from rewetting air-dried soil in short-term laboratory experiments: Original article. Soil Science and Plant Nutrition, 53(2), 181–188. https://doi.org/10.1111/j.1747-0765.2007.00123.x.

Second, the whole manuscript should be edited (not only English) so that readers can understand the research better.

Some minor comments are listed as follows.

L45; lose

Please rephrase the word. Lose is a transitive verb. As an intransitive verb, the word is used as “He lost the game.”

L48; 3) suppressing ammonia oxidation by nitrification bacteria or archaea, known as biological nitrification inhibition (BNI). Brachiaria humidicola and Sorghum bicolor of family Poaceae (also known as Gramineae) are known to release diterpene (brachialactone) and paraquinone derivatives (soligoquinones) respectively from plant roots as their BNI principles [3,4].

How can we know that the purpose of the BNI production is to prevent the leakage of the N from the ecosystem? The role of the BNI may be a defense against pests?

L59; effect

Affect?

L71; despite of

Despite

L74; despite of

Despite

L98; also

Do you need this word?

L99; also

Do you need this word?

L101; In the soil incubation assay, 50 μM of each test compound was exposed to N2O-emitting raw soil of Andisol corn farmland recognized as a N2O emission hotspot.

This part needs citation. For example,

Takeda et al. (2012) Active N2O emission from bacterial microbiota of Andisol farmland and characterization of some N2O emitters, Journal of Basic Microbiology 52(4):477-86. DOI: 10.1002/jobm.201100241

L130; a) As an additional carbon source, 1 mL of 10 mg/mL sucrose solution was supplemented with be final concentration of 0.2%.

The sentence is odd. Please check it. With be final concentration of 0.2%?

L134; b) Alternative sucrose content (ranged from 0.01 to 1.0%) Incubation was done at 25°C in the dark for 7 days.

The sentence is odd. Please check it. What is the subject of the sentence? Alternative sucrose content? Incubation?

L141; N2O accumulated in the headspace (27.3 mL) gas was analysed quantitatively by using an ECD (electron capture detector)-gas chromatography (Shimadzu GC-14B, Kyoto, Japan) column equipped with an electron capture detector (Shimadzu ECD-2014).

What’s the difference between this part and method 2.2? In addition, if authors report the accumulated N2O quantitatively, they need to report the amount of N2O, not the concentration. This is because future research might compare the values with the present research. If only concentrations are reported, they cannot compare the response ratios (response ratio of concentrations are not proportional with response ratio of gas fluxes). Probably a similar experiment by Mori et al. (2016), who measured N2O emissions by 24-hour incubation, can help. They calculated the amount by observing the N2O concentration at 0-hour and 24-hour.

Mori T., Yokoyama D. & Kitayama K. (2016). Contrasting effects of exogenous phosphorus application on N2O emissions from two tropical forest soils with contrasting phosphorus availability. Springer Plus, 1237. https://doi.org/10.1186/s40064-016-2587-5.

L149; for

On?

L174; 3.1. Acceleration of N2O emission in soil bed culture supplemented with nitrogen substrate and sucrose Similar with Winogradsky’s mineral solution-based gellan gum bed, supplementation of nitrogen substrates, particularly NO3, and carbon source such as 0.2% sucrose resulted in active acceleration of N2O emission from the soil bed in the gas-chromatography vial (Fig. 2a) [23]. Although sucrose content ranged from 0.01% to 1.0% were also tested, N2O emission from the soil bed culture was unstable, including control (0% sucrose) (Fig. 2b). Hence, 0.05% sucrose was used for the following experiments.

I could not understand the part. What is the role of this part? Why the impact of 0.2% sucrose addition differ in Fig. 2a and Fig. 2b?

L186; In the N2O emission suppression assay using incomplete denitrifying Pseudomonas spp. isolated from Shinhidaka Andisol corn farm soil [20], N2O emission from 5 g of the soil bed in a gas-chromatography vial and its suppression by Hill reaction-inhibiting herbicides were also tested…

Could not understand. Is the experiment the same as the one described in 2-5? Anyways, it is not the normal writing of a result. Result parts generally does not repeat the method description.

L230; suppression of N2O production

Suppressive impacts on N2O production

L234; suppression of N2O production

Suppressive impacts on N2O production

L238; mean

Please rephrase it. Sounds not scientific.

L239; player

Players

Discussion part.

I could not follow the logic of the discussion part.

For example, the following discussion.

“At 50 μM, two electron-transport inhibiting herbicides, paraquat (1) and isouron (2) exhibited suppression of N2O production from the substrate added soil bed in gas-chromatography vial. Although the effective concentration of the herbicides that is applied to soil particles seems obviously higher level than the residing levels of them in farmland bulk soil, these herbicides practically used showed potent suppression of N2O emission from a model soil bed of the fertilized farmland soil. However, three other electron-transport inhibiting herbicides (3-5) did not show any suppressive effect on N2O emission from the 2 mM NH4NO3 -supplemented soil bed. These results probably mean that both incomplete denitrifiers and nitrifiers are the main player in the farm soil for N2O emission, and some electron transport inhibiting chemicals (1 and 2) indeed exhibited selective steps of nitrification (Fig. 3). Conversely, N2O-producible incomplete denitrification using NO3 − as the substrate was inhibited by 1 and 2 only. Despite a similar pyridilium cation structure, diquat (4) did not show any inhibiting activity against N2O emission in the soil beds.”

In my understanding, sentences before “Conversely” mentioned that paraquat (1) and isouron (2) exhibited suppression of N2O production, but not three other electron-transport inhibiting herbicides (3-5). Then, how about after “Conversely?” N2O-producible incomplete denitrification using NO3 − as the substrate was inhibited by 1 and 2 only. What is mentioned “Conversely?” Are there any contrast sentences? Both parts (before and after “Conversely”) said that 1-2 suppressed N2O but 3-5 did not.

Discussion of the present research is very simple. Please try revising the part so that readers can easily follow the discussion.

Author Response

Please see PDF file attached.

Reviewer 2 Report

Reviewer comments and suggestions

Manuscript ID: applsci-468355

With this paper, authors present a study concerning the Hill reaction inhibitors, like as paraquat, isouron, bromacil, diquat and simazine. These compounds are present in herbicides used in farming activity and they could influence the electron-transporting pathways of nitrate respiration of denitrifying bacteria. N2O in the headspace gas of the cultured vials with inoculates was detected and quantified using ECD gas chromatography.

In my opinion some information should be corrected and completed so that the manuscript can be suitable for the publication in Applied Sciences.

General comments on the whole text:

 Please check the punctuation and spaces in the text.

- Check the text to the end of the line does not leave alone digits or values separated from the unit.

-   Check the text in terms of language, so as to constantly keep one time.

-   The analytical novelty and advantages of the proposed study as compared with previous publications must be strongly supported.

Some concrete comments would be as follows:

- Introduction is well written and presented, but the authors must emphasize the news of this study.

-  More recent bibliographic references are needed.

- The limit of detection and the limit of quantification (LOD and LOQ) are very important instrumental limits, and they should be referred.

- It would be wise to compare the authors' results with other ones available on the specific literature.

- Discussion must be improved, showing how this study could contribute positively to environmental sustainability by focusing on the important subject of nitrogen loss from fertilized soils.

Final comments and considerations: The latest version of this manuscript deserves to be published after the suggestions and corrections listed above are amended.

Author Response

Please see and check our response to the comments, shown in PDF attached.
